# Universal Approximation with Certified Networks

**Maximilian Baader, Matthew Mirman, Martin Vechev**
Department of Computer Science
ETH Zurich, Switzerland
{mbaader,matthew.mirman,martin.vechev}@inf.ethz.ch

## Abstract

Training neural networks to be certifiably robust is critical to ensure their safety against adversarial attacks. However, it is currently very difficult to train a neural network that is both accurate and certifiably robust. In this work we take a step towards addressing this challenge. We prove that for every continuous function $f$, there exists a network $n$ such that: (i) $n$ approximates $f$ arbitrarily close, *and* (ii) simple interval bound propagation of a region $B$ through $n$ yields a result that is arbitrarily close to the optimal output of $f$ on $B$. Our result can be seen as a Universal Approximation Theorem for interval-certified ReLU networks. To the best of our knowledge, this is the first work to prove the existence of accurate, interval-certified networks.

## 1 Introduction

Much recent work has shown that neural networks can be fooled into misclassifying adversarial examples (Szegedy et al., 2014), inputs which are imperceptibly different from those that the neural network classifies correctly. Initial work on defending against adversarial examples revolved around training networks to be empirically robust, usually by including adversarial examples found with various attacks into the training dataset (Gu and Rigazio, 2015; Papernot et al., 2016; Zheng et al., 2016; Athalye et al., 2018; Eykholt et al., 2018; Moosavi-Dezfooli et al., 2017; Xiao et al., 2018). However, while empirical robustness can be practically useful, it does not provide safety guarantees. As a result, much recent research has focused on verifying that a network is certifiably robust, typically by employing methods based on mixed integer linear programming (Tjeng et al., 2019), SMT solvers (Katz et al., 2017), semidefinite programming (Raghunathan et al., 2018a), duality (Wong and Kolter, 2018; Dvijotham et al., 2018b), and linear relaxations (Gehr et al., 2018; Weng et al., 2018; Wang et al., 2018b; Zhang et al., 2018; Singh et al., 2018; Salman et al., 2019).

Because the certification rates were far from satisfactory, specific training methods were recently developed which produce networks that are certifiably robust: Mirman et al. (2018); Raghunathan et al. (2018b); Wang et al. (2018a); Wong and Kolter (2018); Wong et al. (2018); Gowal et al. (2018) train the network with standard optimization applied to an over-approximation of the network behavior on a given input region (the region is created around the concrete input point). These techniques aim to discover specific weights which facilitate verification. There is a tradeoff between the degree of the over-approximation used and the speed of training and certification. Recently, (Cohen et al., 2019b) proposed a statistical approach to certification, which unlike the non-probabilistic methods discussed above, creates a probabilistic classifier that comes with probabilistic guarantees.

So far, some of the best non-probabilistic results achieved on the popular MNIST (Lecun et al., 1998) and CIFAR10 (Krizhevsky, 2009) datasets have been obtained with the simple Interval relaxation (Gowal et al., 2018; Mirman et al., 2019), which scales well at both training and verification time. Despite this progress, there are still substantial gaps between known standard accuracy, experimental robustness, and certified robustness. For example, for CIFAR10, the best reported certified robustness is 32.04% with an accuracy of 49.49% when using a fairly modest $l_\infty$ region with radius 8/255 (Gowal et al., 2018). The state-of-the-art non-robust accuracy for this dataset is $> 95\%$ with experimental robustness $> 50\%$. Given the size of this gap, a key question then is: *can certified training ever succeed or is there a fundamental limit*?

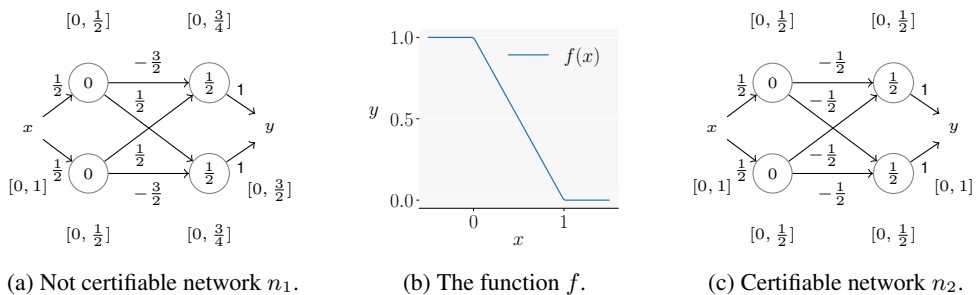

(a) Not certifiable network $n_1$.  (b) The function $f$.  (c) Certifiable network $n_2$.

Figure 2: The ReLU networks $n_1$ (Figure 2a) and $n_2$ (Figure 2c) encode the same function $f$ (Figure 2b). Interval analysis fails certify that $n_1$ does not exceed $[0, 1]$ on $[0, 1]$ while certification succeeds for $n_2$.

In this paper we take a step in answering this question by proving a result parallel to the Universal Approximation Theorem (Cybenko, 1989; Hornik et al., 1989). We prove that for any continuous function $f$ defined on a compact domain $\Gamma \subseteq \mathbb{R}^m$ and for any desired level of accuracy $\delta$, there exists a ReLU neural network $n$ which can certifiably approximate $f$ up to $\delta$ using interval bound propagation. As an interval is a fairly imprecise relaxation, our result directly applies to more precise convex relaxations (e.g., Zhang et al. (2018); Singh et al. (2019)).

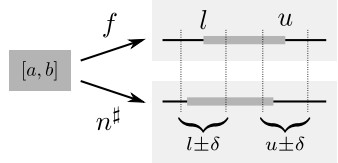

Figure 1: Illustration of Theorem 1.1.

**Theorem 1.1** (Universal Interval-Certified Approximation, Figure 1). Let $\Gamma \subset \mathbb{R}^m$ be a compact set and let $f\colon \Gamma \to \mathbb{R}$ be a continuous function. For all $\delta > 0$, there exists a ReLU network $n$ such that for all boxes $[a, b]$ in $\Gamma$ defined by points $a, b \in \Gamma$ where $a_k \leq b_k$ for all $k$, the propagation of the box $[a, b]$ using interval analysis through the network $n$, denoted $n^\sharp([a, b])$, approximates the set $[l, u] = [\min f([a, b]), \max f([a, b])] \subseteq \mathbb{R}$ up to $\delta$,

$$[l + \delta, u - \delta] \subseteq n^\sharp([a, b]) \subseteq [l - \delta, u + \delta]. \tag{1}$$

We recover the classical universal approximation theorem ($|f(x) - n(x)| \leq \delta$ for all $x \in \Gamma$) by considering boxes $[a, b]$ describing points ($x = a = b$). Note that here the lower bound is not $[l, u]$ as the network $n$ is an approximation of $f$. Because interval analysis propagates boxes, the theorem naturally handles $l_\infty$ norm bound perturbations to the input. Other $l_p$ norms can be handled by covering the $l_p$ ball with boxes. The theorem can be extended easily to functions $f\colon \Gamma \to \mathbb{R}^k$ by applying the theorem component wise.

**Practical meaning of theorem**    The practical meaning of this theorem is as follows: if we train a neural network $n'$ on a given training data set (e.g., CIFAR10) and we are satisfied with the properties of $n'$ (e.g., high accuracy), then because $n'$ is a continuous function, the theorem tells us that there exists a network $n$ which is as accurate as $n'$ and as certifiable with interval analysis as $n'$ is with a complete verifier. This means that if we fail to find such an $n$, then either $n$ did not possess the required capacity or the optimizer was unsuccessful.

**Focus on the existence of a network**    We note that we do not provide a method for training a certified ReLU network – even though our method is *constructive*, we aim to answer an existential question and thus we focus on proving that a given network exists. Interesting future work items would be to study the requirements on the size of this network and the inherent hardness of finding it with standard optimization methods.

**Universal approximation is insufficient**    We now discuss why classical universal approximation is insufficient for establishing our result. While classical universal approximation theorems state that neural networks can approximate a large class of functions $f$, unlike our result, they do not state that robustness of the approximation $n$ of $f$ is actually certified with a scalable proof method (e.g., interval bound propagation). If one uses a non scalable complete verifier instead, then the standard Universal approximation theorem is sufficient.

To demonstrate this point, consider the function $f : \mathbb{R} \to \mathbb{R}$ (Figure 2b) mapping all $x \leq 0$ to 1, all $x \geq 1$ to 0 and all $0 < x < 1$ to $1 - x$ and two ReLU networks $n_1$ (Figure 2a) and $n_2$ (Figure 2c) perfectly approximating $f$, that is $n_1(x) = f(x) = n_2(x)$ for all $x$. For $\delta = \frac{1}{4}$, the interval certification that $n_1$ maps all $x \in [0, 1]$ to $[0, 1]$ fails because $[\frac{1}{4}, \frac{3}{4}] \subseteq n_1^{\sharp}([0, 1]) = [0, \frac{3}{2}] \not\subseteq [-\frac{1}{4}, \frac{5}{4}]$. However, interval certification succeeds for $n_2$, because $n_2^{\sharp}([0, 1]) = [0, 1]$. To the best of our knowledge, this is the first work to prove the existence of accurate, interval-certified networks.

## 2 Related work

After adversarial examples were discovered by Szegedy et al. (2014), many attacks and defenses were introduced (for a survey, see Akhtar and Mian (2018)). Initial work on verifying neural network robustness used exact methods (Katz et al., 2017; Tjeng et al., 2019) on small networks, while later research introduced methods based on over-approximation (Gehr et al., 2018; Raghunathan et al., 2018a; Singh et al., 2018; Salman et al., 2019) aiming to scale to larger networks. A fundamentally different approach is randomized smoothing (Li et al., 2019; Lécuyer et al., 2019; Cohen et al., 2019b), in which probabilistic classification and certification with high confidence is performed.

As neural networks that are experimentally robust need not be certifiably robust, there has been significant recent research on training certifiably robust neural networks (Raghunathan et al., 2018b; Mirman et al., 2018; 2019; Wong and Kolter, 2018; Wong et al., 2018; Wang et al., 2018a; Gowal et al., 2018; Dvijotham et al., 2018a; Xiao et al., 2019; Cohen et al., 2019b). As these methods appear to have reached a performance wall, several works have started investigating the fundamental barriers in the datasets and methods that preclude the learning of a robust network (let alone a certifiably robust one) (Khoury and Hadfield-Menell, 2018; Schmidt et al., 2018; Tsipras et al., 2019). In our work, we focus on the question of whether neural networks are capable of approximating functions whose robustness can be established with the efficient interval relaxation.

**Feasibility Results with Neural Networks**   Early versions of the Universal Approximation Theorem were stated by Cybenko (1989) and Hornik et al. (1989). Cybenko (1989) showed that networks using sigmoidal activations could approximate continuous functions in the unit hypercube, while Hornik et al. (1989) showed that even networks with only one hidden layer are capable of approximating Borel measurable functions.

More recent work has investigated the capabilities of ReLU networks. Here, Arora et al. (2018), based on Tarela and Martínez (1999), proved that every continuous piecewise linear function in $\mathbb{R}^m$ can be represented by a ReLU network. Later, He et al. (2018) reduced the number of neurons needed using ideas from finite elements methods. Relevant to our work, Arora et al. (2018) introduced a ReLU network representations of the $\min$ function. Further, we use a construction method that is similar to the construction for nodal basis functions given in He et al. (2018).

Universal approximation for Lipschitz constrained networks have been considered by Anil et al. (2019) and later by Cohen et al. (2019a). A bound on the Lipschitz constant of a network immediately yields a certified region depending on the classification margin. Anil et al. (2019) proved that the set of Lipschitz networks with the GroupSort activation is dense in the space of Lipschitz continuous functions with Lipschitz constant 1, while Cohen et al. (2019a) provide an explicit construction to obtain the network. We note that both of these works focus on Lipschitz continuous functions, a more restricted class than continuous functions, which we consider in our work.

## 3 Background

In this section we provide the concepts necessary to describe our main result.

**Adversarial Examples and Robustness Verification**   Let $n : \mathbb{R}^m \to \mathbb{R}^k$ be a neural network, which classifies an input $x$ to a label $t$ if $n(x)_t > n(x)_j$ for all $j \neq t$. For a correctly classified input $x$, an adversarial example is an input $y$ such that $x$ is imperceptible from $y$ to a human, but is classified to a different label by $n$.

Frequently, two images are assumed to be "imperceptible" if there $l_p$ distance is at most $\epsilon$. The $l_p$ ball around an image is said to be the adversarial ball, and a network is said to be $\epsilon$-robust around $x$ if

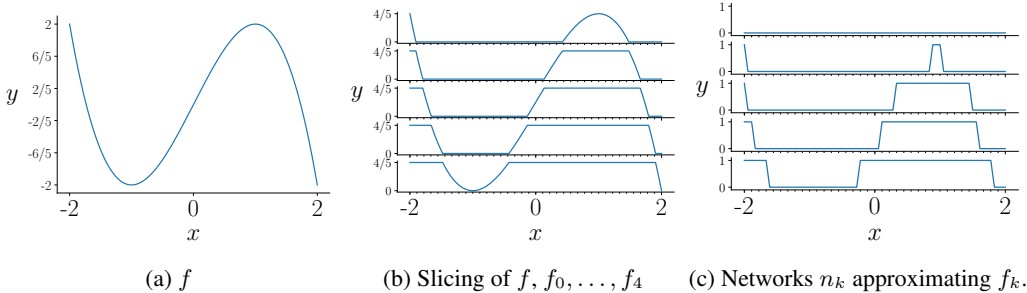

(a) $f$      (b) Slicing of $f, f_0, \dots, f_4$      (c) Networks $n_k$ approximating $f_k$.

Figure 3: Approximating $f$ (Figure 3a) using a ReLU network $n = \xi_0 + \sum_k n_k$. The ReLU networks $n_k$ (Figure 3c) approximate the $N$-slicing of $f$ (Figure 3b), as a sum of local bumps (Figure 6).

every point in the adversarial ball around $x$ classifies the same. In this paper, we limit our discussion to $l_\infty$ adversarial balls which can be used to cover to all $l_p$ balls.

The goal of robustness verification is to show that for a neural network $n$, input point $x$ and label $t$, every possible input in an $l_\infty$ ball of size $\epsilon$ around $x$ (written $\mathbb{B}_\epsilon^\infty(x)$) is also classified to $t$.

**Verifying neural networks with Interval Analysis** The verification technique we investigate in this work is interval analysis. We denote by $\mathcal{B}$ the set of boxes $B = [a, b] \subset \mathbb{R}^m$ for all $m$, where $a_i \leq b_i$ for all $i$. Furthermore for $\Gamma \subseteq \mathbb{R}^m$ we define $\mathcal{B}(\Gamma) := \mathcal{B} \cap \Gamma$ describing all the boxes in $\Gamma$. The standard interval-transformations for the basic operations we are considering, namely $+, -, \cdot$ and the ReLU function $R$ (Gehr et al. (2018), Gowal et al. (2018)) are

$$[a, b] +^\sharp [c, d] = [a + c, b + d] \qquad\qquad -^\sharp [a, b] = [-b, -a]$$
$$R^\sharp([a, b]) = [R(a), R(b)] \qquad\qquad \lambda \cdot^\sharp [a, b] = [\lambda a, \lambda b],$$

where $[a, b], [c, d] \in \mathcal{B}(\mathbb{R})$, and $\lambda \in \mathbb{R}_{\geq 0}$. Furthermore, we used $\sharp$ to distinguish the function $f$ from its interval-transformation $f^\sharp$. To illustrate the difference between $f$ and $f^\sharp$, consider $f(x) := x - x$ evaluated on $x = [0, 1]$. We have $f([0, 1]) = 0$, but $f^\sharp([0, 1]) = [0, 1] -^\# [0, 1] = [0, 1] +^\# [-1, 0] = [-1, 1]$ illustrating the loss in precision that interval analysis suffers from.

Interval analysis provides a sound over-approximation in the sense that for all function $f$, the values that $f$ can obtain on $[a, b]$, namely $f([a, b]) := \{f(x) \mid x \in [a, b]\}$ are a subset of $f^\sharp([a, b])$. If $f$ is a composition of functions, $f = f_1 \circ \cdots \circ f_k$, then $f_1^\sharp \circ \cdots \circ f_k^\sharp$ is a sound interval-transformer for $f$.

Furthermore all combinations $f$ of $+, -, \cdot$ and $R$ are monotone, that is for $[a, b], [c, d] \subseteq \mathcal{B}(\mathbb{R}^m)$ such that $[a, b] \subseteq [c, d]$ then $f^\#([a, b]) \subseteq f^\#([c, d])$ (Appendix A). For boxes $[x, x]$ representing points $f^\sharp$ coincides with $f$, $f^\sharp([x, x]) = f(x)$. This will later be needed.

## 4    PROVING UNIVERSAL INTERVAL-PROVABLE APPROXIMATION

In this section, we provide an explanation of the proof of our main result, Theorem 4.6, and illustrate the main points of the proof.

The first step in the construction is to deconstruct the function $f$ into slices $\{f_k \colon \Gamma \to [0, \frac{\delta}{2}]\}_{0 \leq k < N}$ such that that $f(x) = \xi_0 + \sum_{k=0}^{N-1} f_k(x)$ for all $x$, where $\xi_0$ is the minimum of $f(\Gamma)$. We approximate each slice $f_k$ by a ReLU network $\frac{\delta}{2} \cdot n_k$. The network $n$ approximating $f$ up to $\delta$ will be $n(x) := \xi_0 + \frac{\delta}{2} \sum_k n_k(x)$. The construction relies on 2 key insights, (i) the output of $\frac{\delta}{2} \cdot n_k^\sharp$ can be confined to the interval $[0, \frac{\delta}{2}]$, thus the loss of analysis precision is at most the height of the slice, and (ii) we can construct the networks $n_k$ using local bump functions, such that only 4 slices can contribute to the loss of analysis precision, two for the lower interval bound, two for the upper one.

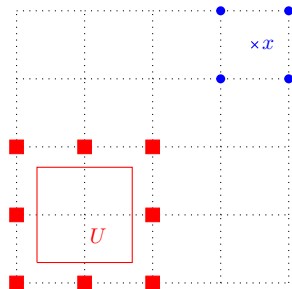
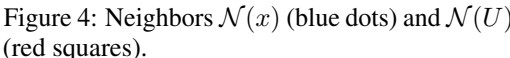

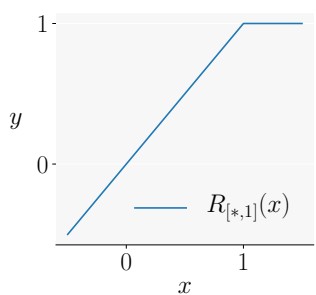

Figure 4: Neighbors $\mathcal{N}(x)$ (blue dots) and $\mathcal{N}(U)$ (red squares).

Figure 5: $R_{[*,b]}(x)$

The slicing $\{f_k\}_{0 \le k < 5}$ of the function $f \colon [-2, 2] \to \mathbb{R}$ (Figure 3a), mapping $x$ to $f(x) = -x^3 + 3x$ is depicted in Figure 3b. The networks $n_k$ are depicted in Figure 3c. In this example, evaluating the interval-transformer of $n$, namely $n^\sharp$ on the box $B = [-1, 1]$ results into $n^\sharp([-1, 1]) = [-2, 6/5]$ lies is within the $\delta = \frac{8}{5}$ bound of $f([-1, 1]) = [-2, 2]$.

**Definition 4.1** (N-slicing (Figure 3b)). Let $\Gamma \subset \mathbb{R}^m$ be a closed $m$-dimensional box and let $f \colon \Gamma \to \mathbb{R}$ be continuous. The *N-slicing* of $f$ is a set of functions $\{f_k\}_{0 \le k < N}$ defined by

$$f_k \colon \Gamma \to \mathbb{R}, \quad x \mapsto \begin{cases} 0 & \text{if } f(x) \le \xi_k, \\ f(x) - \xi_k & \text{if } \xi_k < f(x) < \xi_{k+1}, \\ \xi_{k+1} - \xi_k & \text{if } \xi_{k+1} \le f(x), \end{cases} \quad \forall k \in \{0, \dots, N-1\},$$

where $\xi_k := \xi_0 + \frac{k}{N}(\xi_N - \xi_0)$, $k \in \{1, \dots, N-1\}$, $\xi_0 := \min f(\Gamma)$ and $\xi_N := \max f(\Gamma)$.

To construct a ReLU network satisfying the desired approximation property (Equation (1)) if evaluated on boxes in $\mathcal{B}(\Gamma)$, we need the ReLU network nmin capturing the behavior of $\min$ as a building block (similar to He et al. (2018)). It is given by

$$\text{nmin}(x, y) := \frac{1}{2} \begin{pmatrix} 1 & -1 & -1 & -1 \end{pmatrix} R \left( \begin{pmatrix} 1 & 1 \\ -1 & -1 \\ 1 & -1 \\ -1 & 1 \end{pmatrix} \begin{pmatrix} x \\ y \end{pmatrix} \right).$$

With the ReLU network nmin, we can construct recursively a ReLU network $\text{nmin}_N$ mapping $N$ arguments to the smallest one (Definition A.8). Even though the interval-transformation loses precision, we can establish bounds on the precision loss of $\text{nmin}_N^\sharp$ sufficient for our use case (Appendix A).

Now, we use the clipping function $R_{[*,1]} := 1 - R(1 - x)$ clipping every value exceeding 1 back to 1 (Figure 5) to construct the local bumps $\phi_c$ w.r.t. a grid $G$. $G$ specifies the set of all possible local bumps we can use to construct the networks $n_k$. Increasing the finesse of $G$ will increases the approximation precision.

**Definition 4.2** (local bump, Figure 6). Let $M \in \mathbb{N}$, $G := \{(\frac{i_1}{M}), \dots, \frac{i_m}{M} \mid i \in \mathbb{Z}^m\}$ be a grid, $\ell = 2^{\lceil \log_2 2m \rceil + 1}$ and let $c = \{\frac{i_1^l}{M}, \frac{i_1^u}{M}\} \times \cdots \times \{\frac{i_m^l}{M}, \frac{i_m^u}{M}\} \subseteq G$ be a set of grid points describing the corner points of a hyperrectangle in $G$. We define a ReLU neural network $\phi_c \colon \mathbb{R}^m \to [0, 1] \subset \mathbb{R}$ w.r.t. $G$ by

$$\phi_c(x) := R \left( \text{nmin}_{2m} \bigcup_{1 \le k \le m} \left\{ \begin{array}{l} R_{[*,1]}(M \cdot \ell \cdot (x_k - \frac{i_k^l}{M}) + 1), \\ R_{[*,1]}(M \cdot \ell \cdot (\frac{i_k^u}{M} - x_k) + 1) \end{array} \right\} \right).$$

We will describe later how $M$ and $c$ get picked. A graphical illustration of a local bump for in two dimensions and $c = \{\frac{i_1^l}{M}, \frac{i_1^u}{M}\} \times \{\frac{i_2^l}{M}, \frac{i_2^u}{M}\} = \{c^{ll}, c^{lu}, c^{ul}, c^{uu}\}$ is shown in Figure 6. The local bump $\phi_c(x)$ evaluates to 1 for all $x$ that lie within the convex hull of $c$, namely $\text{conv}(c)$, after which $\phi_c(x)$ quickly decreases linearly to 0. $\phi_c$ has $1 + 2(2d - 1) + 2d$ ReLUs and $1 + \lceil \log_2(2d + 1) \rceil + 1$ layers.

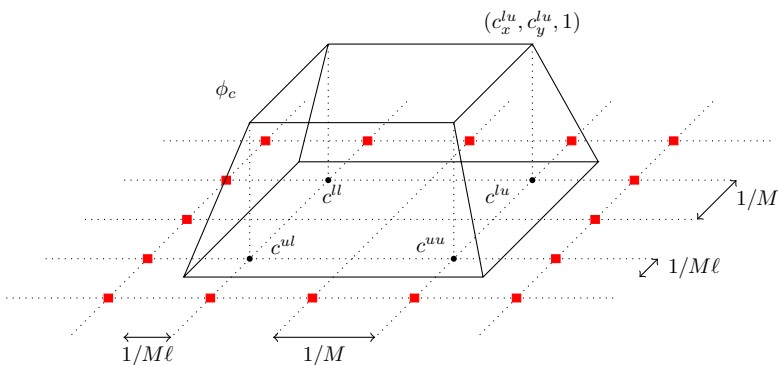

Figure 6: Local bump $\phi_c$, where $c$ contains the points $c^{ll}, c^{lu}, c^{ul}, c^{uu}$. The points in $\mathcal{N}(\text{conv}(c))$ are depicted by the red squares.

By construction $\phi_c(x)$ decreases to 0 before reaching the next neighboring grid points $\mathcal{N}(\text{conv}(c))$, where $\mathcal{N}(x) := \{g \in G \mid ||x - g||_\infty \leq \frac{1}{M}\} \setminus \{x\}$ denotes the neighboring grid points of $x$ and similarly for $\mathcal{N}(U) := \{\mathcal{N}(x) \mid x \in U\} \setminus U$ (Figure 4). The set $\mathcal{N}(\text{conv}(c))$ forms a hyperrectangle in $G$ and is shown in Figure 6 using red squares. Clearly $\text{conv}(c) \subseteq \text{conv}(\mathcal{N}(c))$.

Next, we give bounds on the loss of precision for the interval-transformation $\phi_c^\sharp$. We can show that interval analysis can (i) never produce intervals exceeding $[0, 1]$ and (ii) is precise if $B$ does no intersect $\text{conv}(\mathcal{N}(c)) \setminus \text{conv}(c)$.

**Lemma 4.3.** For all $B \in \mathcal{B}(\mathbb{R}^m)$, it holds that $\phi_c^\sharp(B) \subseteq [0, 1] \in \mathcal{B}$ and

$$\phi_c^\sharp(B) = \begin{cases} [1, 1] & \text{if } B \subseteq \text{conv}(c) \\ [0, 0] & \text{if } B \subseteq \Gamma \setminus \text{conv}(\mathcal{N}(c)). \end{cases}$$

The formal proof is given in Appendix A. The next lemma shows, how a ReLU network $n_k$ can approximate the slice $f_k$ while simultaneously confining the loss of analysis precision.

**Lemma 4.4.** Let $\Gamma \subset \mathbb{R}^m$ be a closed box and let $f : \Gamma \to \mathbb{R}$ be continuous. For all $\delta > 0$ there exists a set of ReLU networks $\{n_k\}_{0 \leq k < N}$ of size $N \in \mathbb{N}$ approximating the $N$-slicing of $f$, $\{f_k\}_{0 \leq k < N}$ ($\xi_k$ as in Definition 4.1) such that for all boxes $B \in \mathcal{B}(\Gamma)$

$$n_k^\sharp(B) = \begin{cases} [0, 0] & \text{if } f(B) \leq \xi_k - \frac{\delta}{2} \\ [1, 1] & \text{if } f(B) \geq \xi_{k+1} + \frac{\delta}{2}. \end{cases} \tag{2}$$

and $n_k^\sharp(B) \subseteq [0, 1]$.

It is important to note that in Equation (2) we mean $f$ and not $f^\sharp$. The proof for Lemma 4.4 is given in Appendix A. In the following, we discuss a proof sketch.

Because $\Gamma$ is compact and $f$ is continuous, $f$ is uniformly continuous by the Heine-Cantor Theorem. So we can pick a $M \in \mathbb{N}$ such that for all $x, y \in \Gamma$ satisfying $||y - x||_\infty \leq \frac{1}{M}$ holds $|f(y) - f(x)| \leq \frac{\delta}{2}$. We then choose the grid $G = (\frac{\mathbb{Z}}{M})^m \subseteq \mathbb{R}^m$.

Next, we construct for every slice $k$ a set $\Delta_k$ of hyperrectangles on the grid $G$: if a box $B \in \mathcal{B}(\Gamma)$ fulfills $f(B) \geq \xi_{k+1} + \frac{\delta}{2}$, then we add a minimal enclosing hyperrectangle $c \subset G$ such that $B \subseteq \text{conv}(c)$ to $\Delta_k$, where $\text{conv}(c)$ denotes the convex hull of $c$. This implies, using uniform continuity of $f$ and that the grid $G$ is fine enough, that $f(\text{conv}(c)) \geq \xi_{k+1}$. Since there is only a finite number of possible hyperrectangles in $G$, the set $\Delta_k$ is clearly finite. The network fulfilling Equation (2) is

$$n_k(x) := R_{[*,1]}\left(\sum_{c \in \Delta_k} \phi_c(x)\right),$$

where $\phi_c$ is as in Definition 4.2. The $n_k$ are depicted in Figure 3c.

Now, we see that Equation (2) holds by construction: For all boxes $B \in \mathcal{B}(\Gamma)$ such that $f \geq \xi_{k+1} + \frac{\delta}{2}$ on $B$ exists $c' \in \Delta_k$ such that $B \subseteq \text{conv}(c')$ which implies, using Lemma 4.3, that $\phi_{c'}^{\sharp}(B) = [1, 1]$, hence

$$n_k^{\sharp}(B) = R_{[*,1]}^{\sharp}(\phi_{c'}^{\sharp}(B) + \sum_{c \in \Delta_k \setminus c'} \phi_c^{\sharp}(B)) \qquad \forall c \neq c' : \phi_c^{\sharp}(B) \subseteq [0,1] \text{(Lemma 4.3)}$$

$$= R_{[*,1]}^{\sharp}([1,1] + [p_1, p_2]) \qquad [p_1, p_1] \in \mathcal{B}(\mathbb{R}_{\geq 0})$$

$$= R_{[*,1]}^{\sharp}([1 + p_1, 1 + p_2])$$

$$= [1, 1].$$

Similarly, if $f(B) \leq \xi_k - \frac{\delta}{2}$ holds, then it holds for all $c \in \Delta_k$ that $B$ does not intersect $\mathcal{N}(\text{conv}(c))$. Indeed, if a $c \in \Delta_k$ would violate this, then by construction, $f(\text{conv}(c)) \geq \xi_{k+1}$, contradicting $f(B) \leq \xi_k - \frac{\delta}{2}$. Thus $\phi_c^{\sharp}(B) = [0, 0]$, and hence $n^{\sharp}(B) = [0, 0]$.

**Theorem 4.5.** Let $\Gamma \subset \mathbb{R}^m$ be a closed box and let $f : \Gamma \to \mathbb{R}$ be continuous. Then for all $\delta > 0$, exists a ReLU network $n$ such that for all $B \in \mathcal{B}(\Gamma)$

$$[l + \delta, u - \delta] \subseteq n^{\sharp}(B) \subseteq [l - \delta, u + \delta],$$

where $l := \min f(B)$ and $u := \max f(B)$.

*Proof.* Pick $N$ such that the height of each slice is exactly $\frac{\delta}{2}$, if this is impossible choose a slightly smaller $\delta$. Let $\{n_k\}_{0 \leq k < N}$ be a series of networks as in Lemma 4.4. Recall that $\xi_0 = \min f(\Gamma)$. We define the ReLU network

$$n(x) := \xi_0 + \frac{\delta}{2} \sum_{k=0}^{N-1} n_k(x). \tag{3}$$

Let $B \in \mathcal{B}(\Gamma)$. Thus we have for all $k$

$$f(B) \geq \xi_{k+2} \Leftrightarrow f(B) \geq \xi_{k+1} + \frac{\delta}{2} \overset{Lemma\ 4.4}{\Rightarrow} n_k^{\sharp}(B) = [1, 1] \tag{4}$$

$$f(B) \leq \xi_{k-1} \Leftrightarrow f(B) \leq \xi_k - \frac{\delta}{2} \overset{Lemma\ 4.4}{\Rightarrow} n_k^{\sharp}(B) = [0, 0]. \tag{5}$$

Let $p, q \in \{0, \ldots, N-1\}$ such that

$$\xi_p \leq l = \min f(B) \leq \xi_{p+1} \tag{6}$$

$$\xi_q \leq u = \max f(B) \leq \xi_{q+1}, \tag{7}$$

as depicted in Figure 7. Thus by Equation (4) for all $k \in \{0, \ldots, p-2\}$ it holds that $n_k^{\sharp}(B) = [1, 1]$ and similarly, by Equation (5) for all $k \in \{q+2, \ldots, N-1\}$ it holds that $n_k^{\sharp}(B) = [0, 0]$. Plugging this into Equation (3) after splitting the sum into three parts leaves us with

$$n^{\sharp}(B) = \xi_0 + \frac{\delta}{2} \sum_{k=0}^{p-2} n_k^{\sharp}(B) + \frac{\delta}{2} \sum_{k=p-1}^{q+1} n_k^{\sharp}(B) + \frac{\delta}{2} \sum_{k=p+1}^{N-1} n_k^{\sharp}(B)$$

$$= \xi_0 + (p-1)[\tfrac{\delta}{2}, \tfrac{\delta}{2}] + \frac{\delta}{2} \sum_{k=p-1}^{q+1} n_k^{\sharp}(B) + [0, 0].$$

Applying the standard rules for interval analysis, leads to

$$n^{\sharp}(B) = [\xi_{p-1}, \xi_{p-1}] + \frac{\delta}{2} \sum_{k=p-1}^{q+1} n_k^{\sharp}(B),$$

where we used in the last step, that $\xi_0 + k\frac{\delta}{2} = \xi_k$. For all terms in the sum except the terms corresponding to the 3 highest and lowest $k$ we get

$$n_k^{\sharp}(B) = [0, 1] \qquad \forall k \in \{p+2, \ldots, q-2\}. \tag{8}$$

| $k$ | slices | $n_k(B)$ |
|---|---|---|
| $N-1$ | ═══ | $[0,0]$ |
| ⋮ | ⋮ | ⋮ |
| $q+2$ | ═══ | $[0,0]$ |
| $q+1$ | ═══ | |
| $q-1$ | ═══ | $u$ |
| $q-2$ | ═══ | $[0,1]$ |
| ⋮ | ⋮ | ⋮ |
| $p+2$ | ═══ | $[0,1]$ |
| $p+1$ | ═══ | |
| $p-1$ | ═══ | $l$ |
| $p-2$ | ═══ | $[1,1]$ |
| ⋮ | ⋮ | ⋮ |
| $0$ | ═══ | $[1,1]$ |

Figure 7: Illustration of the proof for Theorem 4.5.

Indeed, from Equation (6) we know that there is $x \in B$ such that $f(x) \le \xi_{p+1} = \xi_{p+2} - \frac{\delta}{2}$, thus by Lemma 4.4 $n_k^\sharp([x, x]) = [0, 0]$ for all $p + 2 \le k \le q - 2$. Similarly, from Equation (7) we know, that there is $x' \in B$ such that $f(x) \ge \xi_q = \xi_{q-1} + \frac{\delta}{2}$, thus by Lemma 4.4 $n_k^\sharp([x', x']) = [1, 1]$ for all $p + 2 \le k \le q - 2$. So $n_k^\sharp(B)$ is at least $[0, 1]$, and by Lemma 4.4 also at most $[0, 1]$. This leads to

$$n^\sharp(B) = [\xi_{p-1}, \xi_{p-1}] + \frac{\delta}{2} \sum_{k=p-1}^{p+1} n_k^\sharp(B) + \frac{\delta}{2}((q-2)-(p+2)+1)[0,1] + \frac{\delta}{2} \sum_{k=q-1}^{q+1} n_k^\sharp(B)$$

$$= [\xi_{p-1}, \xi_{p-1}] + \frac{\delta}{2} \sum_{k=p-1}^{p+1} n_k^\sharp(B) + [0, \xi_{q-1} - \xi_{p+2}] \qquad + \frac{\delta}{2} \sum_{k=q-1}^{q+1} n_k^\sharp(B).$$

We know further, that if $p + 3 \le q$, than there is an $x \in B$ such that $f(x) \ge \xi_{p+3} = \xi_{p+2} + \frac{\delta}{2}$, hence similar as before $n_{p+1}^\sharp([x, x]) = [1, 1]$ and similarly $n_p^\sharp([x, x]) = [1, 1]$ and $n^\sharp([x, x]) = [1, 1]$. So we know, that $\frac{\delta}{2} \sum_{k=p-1}^{p+1} n_k^\sharp(B)$ includes at least $[3\frac{\delta}{2}, 3\frac{\delta}{2}]$ and at the most $[0, 3\frac{\delta}{2}]$. Similarly, there exists an $x' \in B$ such that $n_{q-1}^\sharp([x', x']) = [0, 0]$, $n_q^\sharp([x', x']) = [0, 0]$ and $n_{q+1}^\sharp([x', x']) = [0, 0]$. This leaves us with

$$[3\tfrac{\delta}{2}, 3\tfrac{\delta}{2}] \subseteq \tfrac{\delta}{2} \sum_{k=p-1}^{p+1} n_k^\sharp(B) \subseteq [0, 3\tfrac{\delta}{2}]$$

$$[0, 0] \subseteq \tfrac{\delta}{2} \sum_{k=q-1}^{q+1} n_k^\sharp(B) \subseteq [0, 3\tfrac{\delta}{2}],$$

If $p + 3 > q$ the lower bound we want to prove becomes vacuous and only the upper one needs to be proven. Thus we have

$$[l + \delta, u - \delta] \subseteq [\xi_{p+2}, \xi_{p-1}] \subseteq n^\sharp(B) \subseteq [\xi_{p-1}, \xi_{q+2}] \subseteq [l - \delta, u + \delta],$$

where $l := \min f(B)$ and $u := \max f(B)$. $\qquad \square$

**Theorem 4.6** (Universal Interval-Provable Approximation). Let $\Gamma \subset \mathbb{R}^m$ be compact and $f : \Gamma \to \mathbb{R}^d$ be continuous. For all $\delta \in \mathbb{R}_{\ge 0}^m$ exists a ReLU network $n$ such that for all $B \in \mathcal{B}(\Gamma)$

$$[l + \delta, u - \delta] \subseteq n^\sharp(B) \subseteq [l - \delta, u + \delta],$$

where $l, u \in \mathbb{R}^m$ such that $l_k := \min f(B)_k$ and $u_k := \max f(B)_k$ for all $k$.

*Proof.* This is a direct consequence of using Theorem 4.5 and the Tietze extension theorem to produce a neural network for each dimension $d$ of the codomain of $f$. $\qquad \square$

Note that Theorem 1.1 is a special case of Theorem 4.6 with $d = 1$ to simplify presentation.

## 5 CONCLUSION

We proved that for all real valued continuous functions $f$ on compact sets, there exists a ReLU network $n$ approximating $f$ arbitrarily well with the interval abstraction. This means that for arbitrary input sets, analysis using the interval relaxation yields an over-approximation arbitrarily close to the smallest interval containing all possible outputs. Our theorem affirmatively answers the open question, whether the Universal Approximation Theorem generalizes to Interval analysis.

Our results address the question of whether the interval abstraction is expressive enough to analyse networks approximating interesting functions $f$. This is of practical importance because interval analysis is the most scalable non-trivial analysis.

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

## A    Proofs for the Universal Interval-Certified Approximation

**Lemma A.1** (Monotonicity). The operations $+, -$ are monotone, that is for all $[a_1, b_1], [a_2, b_2], [c_1, d_1], [c_2, d_2] \in \mathcal{B}(R)$ such that $[a_1, b_1] \subseteq [a_2, b_2]$ and $[c_1, d_2] \subseteq [c_2, d_2]$ holds

$$[a_1, b_1] +^{\sharp} [c_1, d_1] \subseteq [a_2, d_2] +^{\sharp} [c_2, d_2]$$
$$[a_1, b_1] -^{\sharp} [c_1, d_1] \subseteq [a_2, d_2] -^{\sharp} [c_2, d_2]$$
$$[a_1, b_1] \cdot^{\sharp} [c_1, d_1] \subseteq [a_2, d_2] \cdot^{\sharp} [c_2, d_2].$$

Further the operation $*$ and $R$ are monotone, that is for all $[a, b], [c, d] \in \mathcal{B}(R)$ and for all $\lambda \in \mathbb{R}_{\geq 0}$ such that $[a, b] \subseteq [c, d]$ holds

$$\lambda \cdot^{\sharp} [a, b] \subseteq \lambda \cdot^{\sharp} [c, d]$$
$$R^{\sharp}([a, b]) \subseteq R^{\sharp}([c, d]).$$

*Proof.*

$$[a_1, b_1] +^{\sharp} [c_1, d_1] = [a_1 + c_1, b_1 + d_1] \subseteq [a_2 + c_2, b_2 + d_2] = [a_2, d_2] +^{\sharp} [c_2, d_2]$$
$$[a_1, b_1] -^{\sharp} [c_1, d_1] = [a_1 - d_1, b_1 - c_1] \subseteq [a_2 - d_2, b_2 - c_2] = [a_2, d_2] -^{\sharp} [c_2, d_2]$$

$$\lambda \cdot^{\sharp} [a, b] = [\lambda a, \lambda b] \subseteq [\lambda c, \lambda d] = [\lambda c, \lambda d]$$
$$R^{\sharp}([a, b]) = [R(a), R(b)] \subseteq [R(c), R(d)] = R^{\sharp}([c, d]).$$

$\square$

**Definition A.2** ($N$-slicing). Let $\Gamma \subset \mathbb{R}^m$ be a compact $m$-dimensional box and let $f \colon \Gamma \to \mathbb{R}$ be continuous. The $N$-*slicing* of $f$ is a set of functions $\{f_k\}_{0 \leq k \leq N-1}$ defined by

$$f_k \colon \Gamma \to \mathbb{R}, \quad x \mapsto \begin{cases} 0 & \text{if } f(x) \leq \xi_k, \\ f(x) - \xi_k & \text{if } \xi_k < f(x) < \xi_{k+1}, \quad \forall k \in \{0, \ldots, N-1\}, \\ \xi_{k+1} - \xi_k & \text{otherwise,} \end{cases}$$

where $\xi_k := \frac{k}{N}(\xi_{\max} - \xi_{\min})$, $k \in \{0, \ldots, N\}$, $\xi_{\min} := \min f(\Gamma)$ and $\xi_{\max} := \max f(\Gamma)$.

**Lemma A.3** ($N$-slicing). Let $\{f_k\}_{0 \leq k \leq N-1}$ be the $N$-slicing of $f$. Then for all $x \in \Gamma$ we have $f(x) := \xi_0 + \sum_{k=0}^{N-1} f_k(x)$.

*Proof.* Pick $x \in \Gamma$ and let $l \in \{0, \ldots, N-1\}$ such that $\xi_l \leq f(x) \leq \xi_{l+1}$. Then

$$\xi_0 + \sum_{k=0}^{N-1} f_k(x) = \xi_0 + \sum_{k=0}^{l-1} f_k(x) + f_l(x) + \sum_{k=l+1}^{N-1} f_k(x) = \xi_0 + \sum_{k=0}^{l-1} (\xi_{k+1} - \xi_k) + f_l(x)$$
$$= \xi_l + f_l(x) = f(x).$$

$\square$

**Definition A.4** (clipping). Let $a, b \in \mathbb{R}$, $a < b$. We define the *clipping* function $R_{[*, b]} \colon \mathbb{R} \to \mathbb{R}$ by

$$R_{[*, b]}(x) := b - R(b - x).$$

**Lemma A.5** (clipping). The function $R_{[*, b]}$ sends all $x \leq b$ to $x$, and all $x > b$ to $b$. Further, $R_{[*, b]}^{\sharp}([a', b']) = [R_{[*, b]}(a'), R_{[*, b]}(b')]$.

*Proof.* We show the proof for $R_{[a,b]}$, the proof for $R_{[*,b]}$ is similar.

$$x < b \Rightarrow R_{[*,b]}(x) = b - R(b - x) = b - b + x = x$$
$$x \geq b \Rightarrow R_{[*,b]}(x) = b - R(b - x) = b - 0 = b$$

Next,

$$
\begin{aligned}
R^{\sharp}_{[*,b]}([a', b']) &= b -^{\sharp} R^{\sharp}(b -^{\sharp} [a', b']) \\
&= b -^{\sharp} R^{\sharp}(b +^{\sharp} [-b', -a']) \\
&= b -^{\sharp} R^{\sharp}([b - b', b - a']) \\
&= b -^{\sharp} [R(b - b'), R(b - a')] \\
&= b +^{\sharp} [-R(b - a'), -R(b - b')] \\
&= [b - R(b - a'), b - R(b - b')] \\
&= [R_{[*,b]}(a'), R_{[*,b]}(b')].
\end{aligned}
$$

$\square$

**Definition A.6** (nmin). We define the ReLU network nmin: $\mathbb{R}^2 \to \mathbb{R}$ by

$$
\mathrm{nmin}(x, y) := \frac{1}{2} \begin{pmatrix} 1 & -1 & -1 & -1 \end{pmatrix} R\left( \begin{pmatrix} 1 & 1 \\ -1 & -1 \\ 1 & -1 \\ -1 & 1 \end{pmatrix} \begin{pmatrix} x \\ y \end{pmatrix} \right).
$$

**Lemma A.7** (nmin). Let $x, y \in \mathbb{R}$, then $\mathrm{nmin}(x, y) = \min(x, y)$.

*Proof.* Because nmin is symmetric in its arguments, we assume w.o.l.g. $x \geq y$.

$$
\begin{aligned}
\mathrm{nmin}(x, y) &= \frac{1}{2} \begin{pmatrix} 1 & -1 & -1 & -1 \end{pmatrix} R\left( \begin{pmatrix} 1 & 1 \\ -1 & -1 \\ 1 & -1 \\ -1 & 1 \end{pmatrix} \begin{pmatrix} x \\ y \end{pmatrix} \right) \\
&= \frac{1}{2} \begin{pmatrix} 1 & -1 & -1 & -1 \end{pmatrix} R \begin{pmatrix} x + y \\ -x - y \\ x - y \\ -x + y \end{pmatrix}
\end{aligned}
$$

If $x + y \geq 0$, then

$$\mathrm{nmin}(x, y) = \frac{1}{2}(x + y - x + y) = y.$$

If $x + y < 0$, then

$$\mathrm{nmin}(x, y) = \frac{1}{2}(x + y - x + y) = y.$$

$\square$

**Definition A.8** ($\mathrm{nmin}_N$). For all $N \in \mathbb{N}_{\geq 1}$, we define a ReLU network $\mathrm{nmin}_N$ defined by

$$\mathrm{nmin}_1(x) := x$$
$$\mathrm{nmin}_N(x_1, \ldots, x_N) := \mathrm{nmin}(\mathrm{nmin}_{\lceil N/2 \rceil}(x_1, \ldots, x_{\lceil N/2 \rceil}), \mathrm{nmin}_{\lceil N/2 \rceil + 1}(x_{\lceil N/2 \rceil + 1}, \ldots, x_N)).$$

**Lemma A.9.** Let $[a, b], [c, d] \in \mathcal{B}(\mathbb{R})$. Then $\mathrm{nmin}^{\sharp}([a, b], [c, d]) = \mathrm{nmin}^{\sharp}([c, d], [a, b])$ and

$$
\mathrm{nmin}^{\sharp}([a, b], [c, d]) = \begin{cases} [c + \frac{a-b}{2}, d + \frac{b-a}{2}] & \text{if } d \leq a \\ [a + \frac{c-d}{2}, b + \frac{d-c}{2}] & \text{if } a \leq d \text{ and } b < c \\ [a + c - \frac{b+d}{2}, \frac{b+d}{2}] & \text{if } a \leq d \text{ and } b \geq c \end{cases}
$$

*Proof.* The symmetry on abstract elements is immediate. In the following, we omit some of $\sharp$ to improve readability.

$$\text{nmin}^\sharp([a,b],[c,d]) = \frac{1}{2}\begin{pmatrix}1 & -1 & -1 & -1\end{pmatrix} R^\sharp \left(\begin{pmatrix}1 & 1 \\ -1 & -1 \\ 1 & -1 \\ -1 & 1\end{pmatrix}\begin{pmatrix}[a,b] \\ [c,d]\end{pmatrix}\right)$$

$$= \frac{1}{2}\begin{pmatrix}1 & -1 & -1 & -1\end{pmatrix} R^\sharp \left(\begin{pmatrix}[a,b]+[c,d] \\ -[a,b]-[c,d] \\ [a,b]-[c,d] \\ -[a,b]+[c,d]\end{pmatrix}\right)$$

$$= \frac{1}{2}\begin{pmatrix}1 & -1 & -1 & -1\end{pmatrix} R^\sharp \left(\begin{pmatrix}[a+c,b+d] \\ [-b-d,-a-c] \\ [a-d,b-c] \\ [c-b,d-a]\end{pmatrix}\right)$$

$$= \frac{1}{2}\begin{pmatrix}1 & -1 & -1 & -1\end{pmatrix} \begin{pmatrix}[R(a+c),R(b+d)] \\ [R(-b-d),R(-a-c)] \\ [R(a-d),R(b-c)] \\ [R(c-b),R(d-a)]\end{pmatrix}$$

$$= \frac{1}{2}([R(a+c),R(b+d)] - [R(-b-d),R(-a-c)]$$
$$- [R(a-d),R(b-c)] - [R(c-b),R(d-a)])$$

$$= \frac{1}{2}([R(a+c),R(b+d)] + [-R(-a-c),-R(-b-d)]$$
$$+ [-R(b-c),-R(a-d)] + [-R(d-a),-R(c-b)])$$

$$= \frac{1}{2}([R(a+c)-R(-a-c),R(b+d)-R(-b-d)]$$
$$+ [-R(b-c)-R(d-a),-R(a-d)-R(c-b)])$$

Claim: $R(a+c)-R(-a-c) = a+c$. If $a+c > 0$ then $-a-c < 0$ thus the claim in this case. Indeed: If $a+c \leq 0$ then $-a-c \geq 0$ thus $R(a+c) - R(-a-c) = -R(-a-c) = -(-a-c) = a+c$. Similarly $R(b+d) - R(-b-d) = b+d$.

So the expression simplifies to

$$\text{nmin}^\sharp([a,b],[c,d]) = \frac{1}{2}([a+c,b+d] + [-R(b-c)-R(d-a),-R(a-d)-R(c-b)])$$

We proceed by case distinction:

Case 1: $b - c \leq 0$: Then $a \leq b \leq c \leq d$:

$$\text{nmin}^\sharp([a,b],[c,d]) = \frac{1}{2}([a+c,b+d] + [a-d,b-c])$$
$$= \frac{1}{2}([a+c+a-d,b+d+b-c])$$
$$= [a+\tfrac{c-d}{2},b+\tfrac{d-c}{2}]$$

Case 2: $a - d \geq 0$: Then $c \leq d \leq a \leq b$. By symmetry of nmin equivalent to Case 1. Hence

$$\text{nmin}^\sharp([a,b],[c,d]) = [c+\tfrac{a-b}{2},d+\tfrac{b-a}{2}].$$

Case 3: $a - d < 0$ and $b - c > 0$:

$$\text{nmin}^\sharp([a,b],[c,d]) = \frac{1}{2}([a+c,b+d] + [c-b-d+a,0])$$
$$= \frac{1}{2}([a+c+c-b-d+a,b+d])$$
$$= [a+c-\tfrac{b+d}{2},\tfrac{b+d}{2}]$$

Thus we have

$$
\text{nmin}^{\sharp}([a,b],[c,d]) = \begin{cases} [a + \frac{c-d}{2}, b + \frac{d-c}{2}] & \text{if } b \leq c \\ [c + \frac{a-b}{2}, d + \frac{b-a}{2}] & \text{if } d \leq a \\ [a + c - \frac{b+d}{2}, \frac{b+d}{2}] & \text{if } a < d \text{ and } b > c \end{cases}
$$

$\square$

**Definition A.10** (neighboring grid points). Let $G$ be as above. We define the set of *neighboring grid points* of $x \in \Gamma$ by

$$
\mathcal{N}(x) := \{g \in G \mid g \in \|x - g\| \leq \tfrac{1}{M}\} \setminus \{x\}.
$$

For $U \subset \mathbb{R}^m$, we define $\mathcal{N}(U) := \{\mathcal{N}(x) \mid x \in U\} \setminus U$.

**Definition A.11** (local bump). Let $M \in \mathbb{N}$, $G := (\frac{\mathbb{Z}}{M})^m$, $\ell = 2^{\lceil \log_2 2m \rceil + 1}$ and let $c = \{\frac{i_1^l}{M}, \frac{i_1^u}{M}\} \times \cdots \times \{\frac{i_m^l}{M}, \frac{i_m^u}{M}\} \subseteq G$. We define a ReLU neural network $\phi_c \colon \mathbb{R}^m \to [0,1]$ w.r.t. the grid $G$ by

$$
\phi_c(x) := R\left(\text{nmin}_{2m} \bigcup_{1 \leq k \leq m} \left\{ R_{[*,1]}(M\ell(x_k - \tfrac{i_k^l}{M}) + 1), R_{[*,1]}(M\ell(\tfrac{i_k^u}{M} - x_k) + 1) \right\}\right)
$$

**Lemma A.12.** It holds:

$$
\phi_c(x) := \begin{cases} 0 & \text{if } x \notin \text{conv}(\mathcal{N}(c)) \\ 1 & \text{if } x \in \text{conv}(c) \\ \min\left(0, \bigcup_{k=1}^m \{M\ell(x_k - \tfrac{i_k^l}{M}) + 1\} \cup \{M\ell(\tfrac{i_k^u}{M} - x_k) + 1\}\right) & \text{otherwise.} \end{cases}
$$

*Proof.* By case distinction:

- Case $x \notin \mathcal{N}(c)$. Then there exists $k$, such that either $x_k < \frac{i_k^l - 1}{M}$ or $x_k > \frac{i_k^u + 1}{M}$. Then $M\ell(x_k - \frac{i_k^l}{M}) + 1$ or $M\ell(\frac{i_k^u}{M} - x_k) + 1$ is less or equal to 0. Hence

$$
\phi_c(x) = 0.
$$

- Case $x \in \text{conv}(c)$. Then for all $k$ holds $\frac{i_k^l}{M} \leq x_k \leq \frac{i_k^u}{M}$. Thus $M\ell(x_k - \frac{i_k^l}{M}) + 1 \geq 1$ and $M\ell(\frac{i_k^u}{M} - x_k) + 1 \geq 1$ for all k Hence

$$
\phi_c(x) = 1.
$$

  where $\alpha \geq 1$.

- Case otherwise: For all $x$ exists a $k$ such that $M\ell(x_k - \frac{i_k^l}{M}) + 1$ or $M\ell(\frac{i_k^u}{M} - x_k) + 1$ is smaller or equal to all other arguments of the function $\min$ and smaller or equal to 1. If the smallest element is smaller than 0, then $\phi_c(x)$ will evaluate to 0, otherwise it will evaluate to $M\ell(x_k - \frac{i_k^l}{M}) + 1$ or $M\ell(\frac{i_k^u}{M} - x_k) + 1$. Thus we can just drop $R$ and $R_{[*,1]}$ from the equations and take the minimum also over 0:

$$
\phi_c(x) = R\left(\min \bigcup_{k=1}^m \left\{ R_{[*,1]}(M\ell(x_k - \tfrac{i_k^l}{M}) + 1), R_{[*,1]}(M\ell(\tfrac{i_k^u}{M} - x_k) + 1) \right\}\right)
$$

$$
= \min\left(0, \bigcup_{k=1}^m \{(M\ell(x_k - \tfrac{i_k^l}{M}) + 1)\} \cup \{(M\ell(\tfrac{i_k^u}{M} - x_k) + 1)\}\right)
$$

$$
= \min \bigcup_{k=0}^m \{M\ell(x_k - \tfrac{i_k^l}{M}) + 1\} \cup \{M\ell(\tfrac{i_k^u}{M} - x_k) + 1\}
$$

$\square$

**Lemma A.13.** Let $[u_1, 1], \ldots, [u_N, 1]$ be abstract elements of the Interval Domain $\mathcal{B}$. Then

$$\text{nmin}_N^\sharp([u_1, 1], \ldots, [u_N, 1]) = [u_1 + \cdots u_N + 1 - N, 1].$$

*Proof.* By induction. Base case: Let $N = 1$. Then $\text{nmin}_1^\sharp([u_1, 1]) = [u_1, 1]$. Let $N = 2$. Then $\text{nmin}_2^\sharp([u_1, 1], [u_2, 1]) = [u_1 + u_2 - 1, 1]$.

Induction hypothesis: The property holds for $N'$ s.t. $0 < N' \leq N - 1$.

Induction step: Then it also holds for $N$:

$$\begin{aligned}
\text{nmin}_N^\sharp([u_1, 1], \ldots, [u_N, 1]) &= \text{nmin}^\sharp(\text{nmin}_{\lceil N/2 \rceil}^\sharp([u_1, 1], \ldots, [u_{\lceil N/2 \rceil}, 1]), \\
&\qquad \text{nmin}_{N - \lceil N/2 \rceil}^\sharp([u_{\lceil N/2 \rceil + 1}, 1], \ldots, [u_N, 1])) \\
&= \text{nmin}^\sharp([u_1 + \cdots + u_{\lceil N/2 \rceil} + 1 - \lceil N/2 \rceil, 1], \\
&\qquad [u_{\lceil N/2 \rceil + 1} + \cdots u_N + 1 - N + \lceil N/2 \rceil, 1]) \\
&\stackrel{Lemma\ A.9}{=} [u_1 + \cdots + u_N + 2 - \lceil N/2 \rceil - N + \lceil N/2 \rceil - 1, 1] \\
&= [u_1 + \cdots + u_N + 1 - N, 1]
\end{aligned}$$

$\square$

**Lemma A.14.** Let $[a, b], [u, 1] \in \mathcal{B}(\mathbb{R}_{\leq 1})$. Then

$$\text{nmin}^\sharp([a, b], [u, 1]) \subseteq [a + \tfrac{u-1}{2}, \tfrac{b+1}{2}]$$

*Proof.*

$$\text{nmin}^\sharp([a, b], [u, 1]) = \begin{cases} [a + \tfrac{u-1}{2}, b + \tfrac{1-u}{2}] & \text{if } b \leq u \\ [a + u - \tfrac{b+1}{2}, \tfrac{b+1}{2}] & \text{if } b \geq u \end{cases}$$

If $b \leq u$ then $b + \tfrac{1-u}{2} \leq b + \tfrac{1-b}{2} = \tfrac{b+1}{2}$. If $u \leq b$ then $a + u - \tfrac{b+1}{2} \geq a + u - \tfrac{u+1}{2} = a + \tfrac{u-1}{2}$. So

$$\text{nmin}^\sharp([a, b], [u, 1]) \subseteq [a + \tfrac{u-1}{2}, \tfrac{b+1}{2}].$$

$\square$

**Lemma A.15.** Let $N \in \mathbb{N}_{\geq 2}$, let $[u_1, 1], \ldots, [u_{N-1}, 1], [u_N, d] \in \mathcal{B}(\mathbb{R})$ s.t. $b \leq 1$ be abstract elements of the Interval Domain $\mathcal{B}$. Furthermore, let $H(x) := \tfrac{1+x}{2}$. Then there exists a $u \in \mathbb{R}$ s.t.

$$\text{nmin}_N^\sharp([u_1, 1], \ldots, [u_{N-1}, 1], [u_N, d]) \subseteq [u, H^{\lceil \log_2 N \rceil + 1}(d)]$$

*Proof.* By induction: Let $N = 2$:

$$\text{nmin}_2^\sharp([u_1, 1], [u_2, d]) \stackrel{Lemma\ A.14}{=} [a + \tfrac{u_1 - 1}{2}, H(d)]$$

Let $N = 3$:

$$\begin{aligned}
\text{nmin}_3^\sharp([u_1, 1], [u_2, 1], [u_3, d]) &= \text{nmin}^\sharp(\text{nmin}^\sharp([u_1, 1], [u_2, 1]), [u_3, d]) \\
&= \text{nmin}^\sharp([u_1 + u_2 - 1, 1], [u_3, d]) \\
&\subseteq [u_3 + \tfrac{u_1 + u_2 - 2}{2}, H(d)] \\
\text{nmin}_3^\sharp([u_1, 1], [a, b], [u_2, 1]) &= \text{nmin}_3^\sharp([u_3, d], [u_1, 1], [u_2, 1]) \\
&= \text{nmin}^\sharp(\text{nmin}^\sharp([u_3, d], [u_1, 1]), [u_2, 1]) \\
&= \text{nmin}^\sharp([u_3 + \tfrac{u_1 - 1}{2}, H(d)], [u_2, 1]) \\
&\subseteq [u_3 + \tfrac{u_1 + u_2 - 2}{2}, H^2(d)]
\end{aligned}$$

So $\text{nmin}_3^\sharp([u_3, d], [u_1, 1], [u_2, 1])$ is always included in $[u_3 + \tfrac{u_1 + u_2 - 2}{2}, H^2(d)]$.

Induction hypothesis: The statement holds for all $2 \leq N' \leq N - 1$.

Induction step: Then the property holds also for $N$:

$$\text{nmin}_N^\sharp([u_N, d], [u_1, 1], \ldots, [u_{N-1}, 1]) = \text{nmin}^\sharp(\text{nmin}_{\lceil N/2 \rceil}^\sharp([u_N, d], [u_1, 1], \ldots, [u_{\lceil N/2 \rceil - 1}, 1]),$$

$$\text{nmin}_{N - \lceil N/2 \rceil}^\sharp([u_{\lceil N/2 \rceil}, 1], \ldots, [u_{N-1}, 1]))$$

$$= \text{nmin}^\sharp([u', H^{\lceil \log_2 \lceil N/2 \rceil \rceil + 1}(d)], [u'', 1])$$

$$\subseteq \text{nmin}^\sharp([u', H^{\lceil \log_2 N/2 \rceil + 1}(d)], [u'', 1])$$

$$= \text{nmin}^\sharp([u', H^{\lceil \log_2 N - \log_2(2) \rceil + 1}(d)], [u'', 1])$$

$$= \text{nmin}^\sharp([u', H^{\lceil \log_2 N - 1 \rceil + 1}(d)], [u'', 1])$$

$$= \text{nmin}^\sharp([u', H^{\lceil \log_2 N \rceil}(d)], [u'', 1])$$

$$= [u''', H^{\lceil \log_2 N \rceil + 1}(d)]$$

and similarly for other orderings of the arguments. $\qquad\square$

**Lemma A.16.** Let $H(x) := \frac{1+x}{2}$. For all $N \in \mathbb{N}_{>0}$, we have that $d \le 1 - 2^N$ implies $H^N(d) \le 0$.

*Proof.* By induction. $N = 1$: Then $H(1 - 2) = \frac{1+1-2}{2} = 0$

Induction hypothesis. The statement holds for all $N'$ such that $0 < N' \le N$.

Induction step: $N + 1$: $d \le 1 - 2^N$:

$$H^{N+1}(d) \le H^{N+1}(1 - 2^{N+1}) = H^N(H(1 - 2^{N+1})) = H^N(\tfrac{1+1-2^{N+1}}{2}) = H^N(1 - 2^N) \le 0$$

$\qquad\square$

**Lemma A.17.** For all boxes $B \in \mathcal{B}(\mathbb{R}^m)$, we have

$$\phi_c^\sharp(B) = \begin{cases} [1,1] & \text{if } B \subseteq \text{conv}(c) \\ [0,0] & \text{if } B \subseteq \Gamma \setminus \text{conv}(\mathcal{N}(c)) \end{cases}$$

Furthermore, $\phi_c^\sharp(B) \subseteq [0, 1]$.

*Proof.* Let $\phi_c$ be a local bump and let $B = [a, b] \in \mathcal{B}(\mathbb{R}^m)$. Let $[r_k^1, s_k^1], [r_k^2, s_k^2] \in \mathcal{B}(\mathbb{R})$ such that $M\ell([a_k, b_k] - \frac{i_k^l}{M}) + 1 = [r_k^1, s_k^1]$ and $M\ell(\frac{i_k^u}{M} - [a_k, b_k]) + 1 = [r_k^2, s_k^2]$.

- If $[a, b] \subseteq \text{conv}(c)$: Then $1 \le r_k^1$ and $1 \le r_k^2$ for all $k \in \{1, \ldots, m\}$. Thus

$$\phi_c^\sharp([a, b]) = R^\sharp(\text{nmin}_{2m}^\sharp \{R_{[*,1]}^\sharp([r_k^p, s_k^p])\}_{(p,k) \in \{1,2\} \times \{1,\ldots,m\}})$$

$$= R^\sharp(\text{nmin}_{2m}^\sharp \{[1, 1]\}_{(p,k) \in \{1,2\} \times \{1,\ldots,m\}})$$

$$= [1, 1]$$

- If $[a, b] \subseteq \Gamma \setminus \text{conv}(\mathcal{N}(c))$: Then there exists a $(p', k') \in \{1, 2\} \times \{1, \ldots, m\}$ such that $s_{k'}^{p'} \le 1 - 2^{\lceil \log_2 N \rceil + 1}$. Using Lemma A.16 and Lemma A.15, we now that there exists a $u \in \mathbb{R}$ s.t.

$$\phi_c^\sharp([a, b]) = R^\sharp(\text{nmin}_{2m}^\sharp \{R_{[*,1]}^\sharp([r_k^p, s_k^p])\}_{(p,k) \in \{1,2\} \times \{1,\ldots,m\}})$$

$$= R^\sharp(\text{nmin}_{2m}^\sharp \{[R_{[*,1]}(r_k^p), R_{[*,1]}(s_k^p)]\}_{(p,k) \in \{1,2\} \times \{1,\ldots,m\}})$$

$$\subseteq R^\sharp(\text{nmin}_{2m}^\sharp \{[R_{[*,1]}(r_k^p), 1]\}_{(p,k) \ne (p',k')} \cup \{[r_{k'}^{p'}, s_{k'}^{p'}]\})$$

$$\subseteq R^\sharp([u, 0])$$

$$= [0, 0]$$

For any $[a, b] \in \mathcal{B}(\Gamma)$ we have $\phi_c^\sharp([a, b]) \subseteq [0, 1]$ by construction. $\qquad\square$

**Lemma A.18.** Let $\Gamma \subset \mathbb{R}^m$ be a closed box and let $f : \Gamma \to \mathbb{R}$ be continuous. For all $\delta > 0$ exists a set of ReLU networks $\{n_k\}_{0 \leq k \leq N-1}$ of size $N \in \mathbb{N}$ approximating the $N$-slicing of $f$, $\{f_k\}_{0 \leq k \leq N-1}$ ($\xi_k$ as in Definition A.2) such that for all boxes $B \in \mathcal{B}(\Gamma)$

$$n_k^\sharp(B) = \begin{cases} [0, 0] & \text{if } f(B) \leq \xi_k - \frac{\delta}{2} \\ [1, 1] & \text{if } f(B) \geq \xi_{k+1} + \frac{\delta}{2}. \end{cases}$$

and $n_k^\sharp(B) \subseteq [0, 1]$.

*Proof.* Let $N \in \mathbb{N}$ such that $N \geq 2 \frac{\xi_{\max} - \xi_{\min}}{\delta}$ where $\xi_{\min} := \min f(\Gamma)$ and $\xi_{\max} := \max f(\Gamma)$. For simplicity we assume $\Gamma = [0, 1]^m$. Using the Heine-Cantor theorem, we get that $f$ is uniformly continuous, thus there exists a $\delta' > 0$ such that $\forall x, y \in \Gamma. ||y - x||_\infty < \delta' \Rightarrow ||f(y) - f(x)|| < \frac{\delta}{2}$. Further, let $M \in \mathbb{N}$ such that $M \geq \frac{1}{\delta'}$ and let $G$ be the grid defined by $G := (\frac{\mathbb{Z}}{M})^m \subseteq \mathbb{R}^m$.

Let $C(B)$ be the set of corner points of the closest hyperrectangle in $G$ confining $B \in \mathcal{B}(\Gamma)$. We construct the set

$$\Delta_k := \{ C(B) \mid B \in \mathcal{B}(\Gamma) : f(B) \geq \xi_{k+1} + \frac{\delta}{2} \}.$$

We claim that $\{n_k\}_{0 \leq k \leq N-1}$ defined by

$$n_k(x) := R_{[*,1]} \left( \sum_{c \in \Delta_k} \phi_c(x) \right)$$

satisfies the condition.

Case 1: Let $B \in \mathcal{B}(\Gamma)$ such that $f(B) \geq \xi_{k+1} + \frac{\delta}{2}$. Then for all $g \in \mathcal{N}(B)$ holds $f_k(g) = \delta_2$. By construction exists a $c' \in \Delta_k$ such that $B \subseteq \text{conv}(c')$. Using Lemma 4.3 we get

$$n_k^\sharp(B) = R_{[*,1]}^\sharp \left( \sum_{c \in \Delta_k} \phi_c^\sharp(B) \right) = R_{[*,1]}^\sharp \left( \phi_{c'}^\sharp(B) + \sum_{c \in \Delta_k \setminus c'} \phi_c^\sharp(B) \right)$$

$$= R_{[*,1]}^\sharp ([1, 1] + [p_1, p_2]) = [1, 1],$$

where $[p_1, p_2] \in \mathcal{B}(\mathbb{R}_{\geq 0})$. Indeed, by case distinction:

Case 2: Let $B \in \mathcal{B}(\Gamma)$ such that $f(B) \leq \xi_k - \frac{\delta}{2}$. Then for all $g \in \mathcal{N}(B)$ holds $f_k(g) = 0$. Further, $B \cap \text{conv}(\mathcal{N}(c)) = \emptyset$ for all $c \in \Delta_k$ because $G$ is fine enough. Using Lemma 4.3 we obtain

$$n_k^\sharp(B) = R_{[*,1]}^\sharp \left( \sum_{c \in \Delta_k} \phi_c^\sharp(B) \right) = R_{[*,1]}^\sharp ([0, 0]) = [0, 0].$$

By construction we have $n_k^\sharp(B) \subseteq [0, 1]$. □

