# OpenReview forum: "Universal Approximation with Certified Networks"
_ICLR.cc/2020/Conference — Accept (Poster)_

### Official Review · AnonReviewer3 · 2019-10-23
**Official Blind Review #3**

**Rating:** 3

**Review:**

The paper aims to show that there exist neural networks that can be certified by interval bound propagation. It claims to do this by showing that for any function f, there is a neural network g (close to f) such that interval bound propagation on g obtains bounds that are almost as good as one would get by applying interval bound propagation to f.

There are a couple issues with this paper. The first is that the main theorem does not imply the claimed high-level take-away in the paper in any practically relevant regime. This is because while the paper does show that there is a network that approximates f well enough, the size of the network in the construction is exponential (for instance, one of the neural network components in the construction involves summing over all hyperrectangles lying within a grid). The result is only plausibly practically relevant if there is a polynomially-sized network for which the bound propagation works.

The second issue is the comparison to related work, which omits both several key techniques in the literature and two related papers on universal approximation.

On universal approximation papers, see these two: https://arxiv.org/abs/1904.04861 and https://arxiv.org/abs/1811.05381. They consider a different proof strategy but more plausibly yield networks of reasonable size.

On key techniques in the literature: "typically by employing methods based on mixed integerlinear programming, SMT solvers and bound propagation" ignores several major techniques in the field: convex relaxations (SDP + LP), and randomized smoothing. Similarly, "specific training methods have beenrecently developed which aim to produce networks that are certifiably robust" again ignores entire families of techniques; Raghunathan et al. train SDP relaxations to perform well, while Cohen et al. train randomized smoothing to work well. Importantly, randomized smoothing is not about creating an over-approximation of the network but explicitly constructs networks that are smooth.

"some of the best results achieved on the popular MNIST (LeCun et al., 1998) and CIFAR10(Krizhevsky, 2009) datasets have been obtained with the simple Interval approximation (Gowalet al., 2018; Mirman et al., 2019)"
-Is this actually true? I just looked at Mirman et al. and it doesn't seem to explicitly compare to any existing bounds. My impression is that randomized smoothing (Cohen et al.) currently gets the best numbers, if we're allowed to train the network. If not allowed to train the network, Raghunathan et al. (Neurips 2018) performs well and is not compared to in the Mirman paper.

These omissions must be addressed as the introduction and related work is misleading in its current state.

Finally, some writing issues:
>>> While the evidence suggests "no", we prove that for realisticdatasets and specifications, such a network does exist and its certification can beestablished by propagating lower and upper bounds of each neuron through thenetwork

One cannot "prove" something for "realistic datasets", since "realistic datasets" is not a formal assumption. Please fix this.

>>> "the most relaxed yet computationally efficient convex relaxation"

What does most relaxed mean? Most relaxed would most straightforwardly mean outputting the entire space as the set of possibilities at each point, which is clearly not intended.

**Experience Assessment:**

I have published in this field for several years.

**Review Assessment: Checking Correctness Of Derivations And Theory:**

I assessed the sensibility of the derivations and theory.

**Review Assessment: Checking Correctness Of Experiments:**

N/A

**Review Assessment: Thoroughness In Paper Reading:**

I made a quick assessment of this paper.

---

> ### Author Response · Authors · 2019-11-12
> **Answer (1)**
>
> We thank Reviewer 3 for their comments, which we address below.
>
> *On network size from our construction:*
>
> The focus of this work was to present a proof which establishes the existence of interval certifiable networks; our focus was not on an efficient construction.
>
> Nevertheless, we do not sum over all boxes in the Grid, as Reviewer 3 claims; we sum over all boxes fulfilling $f(B) \geq \xi_{k+1}+ \delta/2$. One can instantly obtain a more efficient construction by modifying the construction of the set $\Delta_k$ slightly: Instead of adding all boxes $B$ fulfilling $f(B) \geq \xi_{k+1}+ \delta/2$, we only add the largest box $B’$ containing $B$, fulfilling $f(B’) \geq \xi_{k+1}+ \delta/2$ (if there are multiple largest boxes, pick any one). This drastically reduces the number of neurons. We leave the refinement of this heuristic as future work.
>
> *Related work [1] and [2] consider more restricted functions than us and are either non-constructive or are worst-case exponential in the dimension:*
>
> Reviewer 3 claims works [1] and [2] are related and produce networks of more reasonable sizes than our work.
>
> First, our work considers a more general class of functions than [1] and [2]: we consider any continuous function, while [1] and [2] considers the more restricted class of Lipshitz-continuous functions.
>
> Second, despite the fact [1] and [2] consider more restricted functions, the proof of [1] seems to actually by non-constructive, while the proof of [2] produces networks of size that is exponential in the worst-case, like we do.
>
> To elaborate more:
> - The proof given in [1] does not seem to be constructive: [1] shows that the set of Lipschitz networks with the GroupSort activation function are dense in the space of Lipschitz functions.
> - While the construction given in [2] is constructive, its worst case complexity is also exponential in the input dimension $m$:
>     - Consider the piecewise linear function $f \colon [0,1]^m -> [0,1]$, mapping the center point of $[0,1]^m$ to 1, from
>       which $f$ decreases in a piecewise linear fashion to 0 at the boundary of $[0,1]^m$. It can be easily seen, that $f$ is
>       made up out of $2m$ affine functions in the sense of Lemma 1 in [2].
>     - Next, we define the function $g \colon [0,d]^m -> [0,1]$, mapping the center points of the cubes defined by the integer
>       grid to 1, while $g$ decreases in a piecewise linear fashion to 0 at the boundary of every integer cube. We see that $g$
>       is made up out of $2m * d^m$ affine functions in the sense of Lemma 1 in [2].
>     - The width of the first hidden layer of the 3 layer FullSort network in Equation (11) in [2] is equal to the number of affine
>       functions, that is in our case for $g$, the width is $2m * d^m$ which is exponential in $m$ (input dimension).
>
> Further, when it comes to network depth, the two methods are not comparable. That is, if we compare the depth of the network in [2] with $3 + \lceil log_2(2m+1) \rceil$ from our work, we observe that the depth is non-comparable, because [2] uses 2 FullSort layers while we use ReLU layers. Using the FullSort activation function, we could immediately reduce the number of layers to 1 FullSort layer followed by 2 ReLU layers by replacing the $nmin_{2m}$ and $R_[*,1]$ in the construction of the local bump functions with the FullSort activation.
>
> Please note that there are cases where our construction yields significantly smaller networks compared to the one constructed in [2]. Considering the function $g$ from above and $\delta = 2$, our construction would return the network mapping all inputs to 0, while the construction of [2] would return a network of width $2m * d^m$.
>
> [1] Anil et. al.: https://arxiv.org/abs/1811.05381
> [2] Jeremy E. Cohen: https://arxiv.org/abs/1904.04861
> [3] Jeremy M. Cohen et. al.: http://proceedings.mlr.press/v97/cohen19c.html
> [4] Jeremy M. Cohen: https://openreview.net/forum?id=SJxSDxrKDr&noteId=HJlVRSuADr
> [5] Zhang et. al.: arXiv preprint arXiv:1906.06316
> [6] https://github.com/eth-sri/diffai
> [7] Raghunathan et al.: http://papers.nips.cc/paper/8285-semidefinite-relaxations-for-certifying-robustness-to-adversarial-examples
> [8] Gowal et. al.: https://arxiv.org/abs/1810.12715
> [9] Wong et. al.: http://papers.nips.cc/paper/8060-scaling-provable-adversarial-defenses

---

> > ### Comment · AnonReviewer3 · 2019-11-12
> > **On network size**
> >
> > Thank you for your response. Can you clarify what is the best upper bound on network size that can be given? You say "This drastically reduces the number of neurons", but it is not clear to me what the new result should be. It sounds from the rest of this response that it should still be exponential?
> >
> > A related question would be what network size you believe this approach should give (independently of what you can currently prove).
> >
> > > we consider any continuous function, while [1] and [2] considers the more restricted class of Lipshitz-continuous functions.
> >
> > Is this really a meaningful distinction? On any compact set Lipschitz functions are dense in the continuous functions. The fact that [1] and [2] are also exponential seems like the more relevant point here.

---

> > > ### Author Response · Authors · 2019-11-12
> > > **Answer**
> > >
> > > Thank you for the questions.
> > >
> > > Q: What is the best upper bound on network size that can be given? What do you think is the best upper bound that could be achieved with this approach? You say "This drastically reduces the number of neurons", but it is not clear to me what the new result should be. It sounds from the rest of this response that it should still be exponential?
> > >
> > > A: In the general case, if we pick a generic continuous function and a small enough $\delta$, then our construction is exponential in the input dimension. The heuristic we mention reduces the number of local bumps, however, the construction still stays exponential in the general case.
> > >
> > > If we work with specific functions and values of $\delta$, then one may obtain better bounds. For example, if we consider the same function $g \colon [0,d]^m \to [0,1]$ defined in our previous reply together with $\delta=2$, then we do not need any neurons in hidden layers to provably approximate the function.
> > >
> > > Q: The fact that [1] and [2] are also exponential seems like the more relevant point here.
> > >
> > > A: Yes, we agree that the exponential scaling of [2] is the most relevant point here. The scaling of [1] is hard to establish, as it does not provide an explicit construction of the neural network.
> > >
> > > Q: Is there really a meaningful distinction between considering continuous and Lipschitz-continuous functions as on any compact set Lipschitz functions are dense in the continuous functions?
> > >
> > > A: We believe it is a meaningful distinction to make. The reason is that our theorem handles continuous functions directly and has the advantage of certifying arbitrary precise networks with arbitrary small error $\delta$. To see the advantage, consider function $\sqrt{x}$ for $x \in [0,1]$ (continuous but not Lipschitz continuous) approximated with Lipschitz functions. Here, the Lipschitz constant of the approximation tends to infinity as the approximation improves. This implies that the error $\delta$ gets arbitrarily large (in the language of [2], certifiable regions get arbitrarily small).
> > >
> > > Finally, we note that we focus on a different certification methodology than [1] and [2]: [1] and [2] focus on certification via Lipschitz constants, while we focus on certification via linear relaxations. We are happy to clarify this point better in the paper.

---

> ### Author Response · Authors · 2019-11-12
> **Answer (2)**
>
> *On Randomised Smoothing vs. Linear Relaxation Guarantees*
>
> We updated the paper to clarify the relationship to randomised smoothing in the paper. In randomised smoothing, the classifier is a smoothed version of a base classifier (e.g. a neural network). It is infeasible to evaluate the smoothed classifier exactly. An approximated version of the smoothed classifier is constructed using Monte Carlo estimation. Thus the classifier is *not* a neural network, but a randomly sampled neural network. The certification is not given for a neural network, but for a randomly sampled one.
>
> Further, contrary to neural networks, the classification is probabilistic (with high confidence). Similarly, certification is also probabilistic (with high confidence). While we think that randomised smoothing is a promising idea, the guarantees provided by the two methods differ.
>
> For further discussion on comparing the guarantees provided by smoothing with linear relaxations, please refer to a recent comment [4] by Jeremy M. Cohen (author of the randomized smoothing paper [3]).
>
> *Regarding state-of-the-art results on MNIST and CIFAR10:*
>
> Considering CIFAR10 with $l_\infty$ perturbations up to 2/255, Gowal et. al. [8] report an accuracy of 70.1% with certified robustness of 50% while Wong et. al. [9] report an accuracy of 68.3% with reported accuracy of 53.9%. Considering $l_\infty$ perturbations up to 8/255, Mirman et. al. report an accuracy of 46.2% with certified robustness of 27.2% (see [6]) while Zhang et al. [5] report an accuracy of 41.3% with certified robustness of 28.8%.
> While the work of Raghunathan et. al. [7] is very interesting, it does not scale to networks of the size needed to achieve state-of-the-art certification and accuracy.
>
> [1] Anil et. al.: https://arxiv.org/abs/1811.05381
> [2] Jeremy E. Cohen: https://arxiv.org/abs/1904.04861
> [3] Jeremy M. Cohen et. al.: http://proceedings.mlr.press/v97/cohen19c.html
> [4] Jeremy M. Cohen: https://openreview.net/forum?id=SJxSDxrKDr&noteId=HJlVRSuADr
> [5] Zhang et. al.: arXiv preprint arXiv:1906.06316
> [6] https://github.com/eth-sri/diffai
> [7] Raghunathan et al.: http://papers.nips.cc/paper/8285-semidefinite-relaxations-for-certifying-robustness-to-adversarial-examples
> [8] Gowal et. al.: https://arxiv.org/abs/1810.12715
> [9] Wong et. al.: http://papers.nips.cc/paper/8060-scaling-provable-adversarial-defenses

---

### Official Review · AnonReviewer2 · 2019-10-23
**Official Blind Review #2**

**Rating:** 8

**Review:**

Summary:
This paper proves a theoretical result, that for every continuous function f, there exists a ReLU network approximating f well, and for which interval propagation will result in tight bounds.

The proof is by construction: the function to approximate is decomposed into a sum of functions (slices) with bounded range, which will be approximable to the correct precision. Each slices is approximated as a sum of local bumps, who essentially approximates an indicator function over a grid.

Comments:
* Equation (1) of Theorem 1.1, I assume that the left inclusion could be replaced by [l, u], given that interval propagation generate valid bounds? Or is the case that due to the approximating network being only an approximation, interval propagation on it might not give actual bounds on the real function?

Opinion:
The paper is very technical and takes some effort to parse to understand the construction, but the authors do a good job of providing examples and illustrations to make it at least intuitive. The content is clearly novel and the result is far from trivial.

The only thing that i'm slightly wary is the significance of what is proven. The paper shows that approximating a continuous function with a network such that bound propagation works well on it is feasible.
On the other hand, the motivation given is "given a dataset and a specification, is there a network that is both certified and accurate with respect to these", which is a slightly different problem. If this was proven to be yes, then we would know that when our robust training fails, this is due to optimization not being good enough or our not network not having enough capacity. The result proven here does not suffice, because it pre-suppose the existence of a correct function, that would match the dataset and specification, which is not guaranteed (see for example the proofs that robustness and accuracy are at odds, or you can just imagine verification of robustness for too-large radius which would lead to infeasible specification). I still like the paper and I think it might just be a matter of framing the abstract and introduction.

Do the authors have an implementation, even for only toy-size examples, of the generation of a Neural Network approximating a given function or is this purely theoretical? Especially when the proof is constructive like this, this can strengthen the confidence that no subtle mistakes have been made. If so, this should be mentioned.

Potential typos:
Page 4: "Further" -> "Furthermore"?
Page 5: "Increasing the fines" -> "Increasing the finesse?"
Page 6, just before Lemma 4.3, missing a mathcal before the N?

Disclaimer:
I carefully checked the derivations in the paper but only assessed the sensibility of the proofs in the appendix.


**Experience Assessment:**

I have published one or two papers in this area.

**Review Assessment: Checking Correctness Of Derivations And Theory:**

I assessed the sensibility of the derivations and theory.

**Review Assessment: Checking Correctness Of Experiments:**

N/A

**Review Assessment: Thoroughness In Paper Reading:**

N/A

---

> ### Author Response · Authors · 2019-11-12
> **Answer**
>
> We thank Reviewer 2 for their comments, suggestions and in particular for carefully checking the derivations. We reframed the Abstract and Introduction as suggested and added a small paragraph on what the theorem means in the case of a given dataset.
>
> Q: Can the left inclusion of Equation (1) in Theorem 1.1 be replaced by $[l, u]$ or is $[l + \delta, u - \delta]$ needed because the network $n$ is an approximation of the function $f$?
> Yes, we need the lower bound to be $[l+\delta, u-\delta]$ because the network $n$ is an approximation of $f$. We clarified this point in the paper.
>
> Q: Do you provide an implementation following the construction of your theorem?
> Yes, we provide a Python implementation here [ https://glot.io/snippets/fhrl5ouae3 ], and a Haskell implementation here [ https://glot.io/snippets/fhrl8jt8pp ].

---

### Official Review · AnonReviewer1 · 2019-10-25
**Official Blind Review #1**

**Rating:** 6

**Review:**

This paper proves the universal approximation property of interval-certified ReLU networks. Such a network achieves the best certified accuracy against l_infty adversaries at large epsilons on MNIST and CIFAR10, but the certified accuracy is still far from satisfaction. With the results from this paper, we can at least make sure the interval-certified ReLU networks is able to fit the robust classifier with arbitrarily small error if it exists.

I did not check the details of the proofs, but I think such a theorem is a great reference for future work.

However, we cannot conclude whether a network with a moderate size exists to fit the robust classifier well from this paper. From the construction of local bumps (nmin), the number of neurons needed to represent a bump function grows exponentially with the input dimension, which makes the size of the network vacuously large when the input is images. We are not getting any closer to improving the certified robustness with this paper.

**Experience Assessment:**

I have read many papers in this area.

**Review Assessment: Checking Correctness Of Derivations And Theory:**

I assessed the sensibility of the derivations and theory.

**Review Assessment: Checking Correctness Of Experiments:**

N/A

**Review Assessment: Thoroughness In Paper Reading:**

I made a quick assessment of this paper.

---

> ### Author Response · Authors · 2019-11-12
> **Answer**
>
> We thank Reviewer 1 for their comments. We would like to clarify, that the network needed to represent a bump function consists of $6m-1$ ReLUs grouped in $2 + \lceil \log_2(2m+1) \rceil $ layers. Thus, the number of neurons needed to represent a bump function does not grow exponentially with the input dimension $m$. Nevertheless, Reviewer 1 is right that the worst case scaling of the size network with its current construction is exponential. In the comment for Reviewer 3, we provide a (straightforward) improvement. However, in the worst case, the number of bump functions still grows exponentially with the input dimension. Further discussion is given in the comment to Reviewer 3 [ https://openreview.net/forum?id=B1gX8kBtPr&noteId=Hyg928YvoH ].

---

### Author Response · Authors · 2019-10-07
**Clarification**

To be fully clear, we note that B := [a,b] for Theorem 1.1.

---

### Author Response · Authors · 2019-11-12
**General comment on the updated paper:**

We would like to thank the reviewers for the feedback and comments. We incorporated the suggested changes and uploaded the updated version of the paper. In particular, we changed Abstract, Introduction and Related Work to improve overall presentation and clarity. We
- added clarifications and reframed the abstract and intro as suggested by Reviewer 2,
- provide both, Haskell [ https://glot.io/snippets/fhrl8jt8pp ] and Python [ https://glot.io/snippets/fhrl5ouae3 ] code of the construction which can be run directly,
- added the clarifications and citations suggested by Reviewer 3.

---

### Decision · Program_Chairs · 2019-12-19

**Decision:**

Accept (Poster)

**Comment:**

This work shows that there exist neural networks that can be certified by interval bound propagation. It provides interesting and surprising theoretical insights, although analysis requires the networks to be impractically large and hence does not directly yield practical advances.